# Targeting the Tumor Vascular Supply to Enhance Radiation Therapy Administered in Single or Clinically Relevant Fractionated Schedules

**DOI:** 10.3390/ijms25158078

**Published:** 2024-07-24

**Authors:** Michael R. Horsman

**Affiliations:** Experimental Clinical Oncology-Department of Oncology, Aarhus University Hospital, DK-8200 Aarhus, Denmark; mike@oncology.au.dk; Tel.: +45-78454973

**Keywords:** combretastatin A-1 phosphate, CA1P/OXi4503, vascular disrupting agents, radiation, C3H mammary carcinoma, tumor growth delay, local tumor control

## Abstract

This pre-clinical study was designed to demonstrate how vascular disrupting agents (VDAs) should be administered, either alone or when combined with radiation in clinically relevant fractionated radiation schedules, for the optimal anti-tumor effect. CDF1 mice, implanted in the right rear foot with a 200 mm^3^ murine C3H mammary carcinoma, were injected with various doses of the most potent VDA drug, combretastatin A-1 phosphate (CA1P), under different schedules. Tumors were also locally irradiated with single-dose, or stereotactic (3 × 5–20 Gy) or conventional (30 × 2 Gy) fractionation schedules. Tumor growth and control were the endpoints used. Untreated tumors had a tumor growth time (TGT5; time to grow to 5 times the original treatment volume) of around 6 days. This increased with increasing drug doses (5–100 mg/kg). However, with single-drug treatments, the maximum TGT5 was only 10 days, yet this increased to 19 days when injecting the drug on a weekly basis or as three treatments in one week. CA1P enhanced radiation response regardless of the schedule or interval between the VDA and radiation. There was a dose-dependent increase in radiation response when the combined with a single, stereotactic, or conventional fractionated irradiation, but these enhancements plateaued at around a drug dose of 25 mg/kg. This pre-clinical study demonstrated how VDAs should be combined with clinically applicable fractionated radiation schedules for the optimal anti-tumor effect, thus suggesting the necessary pre-clinical testing required to ultimately establish VDAs in clinical practice.

## 1. Introduction

An essential requirement for the growth and development of solid tumors is a functional blood supply [1,2]. Typically, tumors form their own vascular system, from the normal host blood vessels, by the process of angiogenesis [3,4]. The significance of this tumor neo-vascularization makes it a potential therapeutic target and two key approaches have evolved [5,6]. One, first proposed in the 1970s, involves controlling the development of the new vascular network by inhibiting the angiogenesis process [7,8,9]. The other method involves compromising the function of the blood vessels that have already developed at the time of diagnosis [10,11]. This latter approach is often considered a more recent development, yet the concept of specifically targeting the established tumor vasculature as a potential therapeutic approach was actually proposed as early as the 1920s [12], and the first vascular disrupting agent (VDA), colchicine, was tested in both animal models and cancer patients a decade later [13,14]. Studies have since shown that various physical-based treatments (i.e., hyperthermia and photodynamic therapy), biological response modifiers or cytokines (i.e., tumor necrosis factor and interleukins), and established chemotherapeutic agents (i.e., vinka alkaloids and arsenic trioxide) can induce vascular damage [15,16]. Consequently, treatments designed to specifically damage tumor vasculature have been developed. These include various ligand-based approaches that use antibodies, peptides, or growth factors that selectively bind to tumor vessels [15,16]. They also include small-molecule drugs, of which there are two main types. One group is the flavonoids (i.e., flavone acetic acid and its derivative dimethylxanthenone-4-acetic acid), which induce cytokines, especially tumor necrosis factor alpha, causing extensive tumor necrosis, as a result of the vascular collapse [17]. The other group is tubulin binding agents (i.e., combretastatin A-4 phosphate, ZD6126, the phosphate prodrug of N-acetyl-colchinol, and plinabulin), which target proliferating endothelial cells causing cell-shape changes and detachment, ultimately leading to vascular collapse and the induction of widespread necrosis [18].

The lead tubulin-binding agent is generally considered to be combretastatin A-4 phosphate (CA4P), a chemical originally derived from the bark of the African Bush Willow tree (Combretum caffrum) due to its ability to induce significant vascular damage at relatively non-toxic doses [19]. A more potent, non-toxic, derivative is combretastain A-1 phosphate (CA1P/OXi4503). It is superior to CA4P due to its ability to induce greater vascular disruption [20,21] and its oxidative activation to a quinine intermediate, thereby giving it additional direct cell-killing properties [22]. It is well established that vascular targeting agents have no clinical relevance if used as a stand-alone therapy [15,16]. For their true clinical potential to be realized they must be combined with more conventional therapies, especially radiation. However, any agent that targets tumor vasculature has the potential to modify the tumor microenvironment and increase the adverse microenvironmental conditions of oxygen and nutrient deprivation [15]. Such changes could result in an increase in the level of radiation-resistant tumor hypoxia, which would have the potential to reduce the efficacy of any subsequent radiation treatment [23]. Understanding how to combine VDAs with radiation, especially using clinically relevant fractionated radiation schedules, is clearly necessary for their ultimate clinical implementation. The aim of our current study was to use our established murine C3H mammary carcinoma model to undertake a comprehensive investigation into the dose- and time-dependent effects of combining a VDA with radiation for the greatest anti-tumor activity. To demonstrate this effect, we used the current leading VDA (CA1P) and clinically relevant radiation treatments.

## 2. Results

Examples of the growth of this C3H mammary carcinoma under control conditions or following treatment with different CA1P schedules is shown in Figure 1A. Control, untreated tumors grew exponentially from 200 to 1000 mm^3^ and had a mean (±1 SEM) TGT5 of 6.2 days (5.8–6.6). This growth was significantly delayed following a single injection of CA1P (50 mg/kg), with the TGT5 significantly increased to 8.7 days (8.3–9.1). Administering the drug more often further delayed the tumor growth. Figure 1B shows the effect of varying the CA1P dose on TGT5, under the different treatment schedules. Compared to untreated controls, single drug treatments showed a dose-dependent increase in TGT5 but only up to 10 mg/kg, with higher drug doses only producing a small additional increase in TGT5, reaching 9.5 days (8.6–10.4) at the highest drug dose shown (100 mg/kg). This effect on TGT5 could be substantially increased by repeatedly administering the CA1P on a weekly basis, and here the effect definitely increased with higher drug doses. Compared to a weekly schedule, a slightly larger effect was obtained if the maximum number of drug treatments (three) was given during a one-week period, although this difference was lost at the highest drug doses.

Time- and dose-dependent effects of CA1P on radiation response were also investigated and the results are shown in Figure 2. The TGT5 for radiation alone (10 Gy) was found to be 13.2 days (12.6–13.7) and this was significantly increased by combining with CA1P (Figure 2A). However, the enhancement appeared to be generally independent of the timing and sequence between CA1P and radiation, although there is a suggestion that the combination effect may have been slightly greater than the simple additive effect of each treatment alone. This possible additional benefit of combining CA1P and radiation is also suggested where the CA1P dose was varied (Figure 2B).

The drug’s ability to enhance radiation response is clearly also seen when using the local tumor control endpoint (Figure 3A). Here the TCD50 value (with 95% confidence intervals) of 53 Gy (51–55) for radiation alone was significantly decreased to 41 (38–45) when a single dose of CA1P (50 mg/kg) was injected 30 min after irradiating. These TCD50 values, along with other values obtained with different drug radiation intervals are summarized in Figure 3B and again confirm the general lack of a schedule dependency, although giving the drug 24 h prior to irradiating may have been slightly more beneficial.

The effect of CA1P on more clinically relevant fractionated radiation schedules using the tumor control endpoint are shown in Figure 4 and Figure 5. Treating tumors with 3 × 15 Gy resulted in a clamped top-up TCD50 dose of 33 Gy (24–46), which was significantly reduced to 12 Gy (8–19) if mice were treated with 10 mg/kg CA1P after each of the 3 × 15 Gy irradiations (Figure 4A). The effect of varying the three irradiation doses from 5 to 20 Gy on the TCD50 values and the influence of using different CA1P doses is shown in Figure 4B. Increasing the radiation dose/fraction clearly enhanced tumor response. Although there was some variability in response following the addition of the various drug doses, the trends suggest a small enhancement with a 5 mg/kg dose, a larger effect with 10 mg/kg, but little or no additional benefit of using a higher 25 mg/kg drug dose. The results obtained using a conventional 30 × 2 Gy are shown in Figure 5. There the clamped top-up TCD50 value for radiation only was 59 Gy (48–47). This was significantly reduced to 52 Gy (48–57) and 47 Gy (42–53) when mice received respective drug doses of 10 or 25 mg/kg, 1 h after the last irradiation each week (i.e., fractions 10, 20, and 30).

## 3. Discussion

This study was designed to demonstrate how VDAs should be combined with radiotherapy to improve outcome, especially when using clinically relevant fractionated schedules. The tumor model used for these studies was our established C3H mammary carcinoma model. We have previously used this model to test the efficacy of a range of VDAs [15]. Some preliminary studies with CA1P have even been published using this model [21,24,25]. It is also a model in which measurements of VDA activity have been correlated with clinical estimates [26]. Furthermore, it is one of the few models in which the most clinically relevant endpoint of local tumor control can be assessed. Using that particular endpoint, this model has been instrumental in establishing the criteria for several clinical trials with radiation sensitizers [27,28].

The VDA chosen for this proof-of-principal study was CA1P, probably the most potent VDA developed. This drug has dual functional activity in that it can induce vascular damage, like its parent compound CA4P, as well as having a direct cytotoxic effect [20,21,22]. In fact, in this C3H mammary carcinoma model, we have previously shown similar perfusion-induced decreases following treatment with similar doses of CA4P and CA1P, yet a far greater drug dose-dependent effect of CA1P on the development of necrosis and tumor growth inhibition [21]. Other pre-clinical studies have reported CA1P to have superior anti-tumor effects compared to CA4P [20,29,30,31,32]. Most studies with CA1P used only single drug dose treatments [29,30,31,33,34,35,36,37,38]. Some also included multiple-drug dosing [20,32,39,40,41,42]. Several studies even combined CA1P with other drugs, including cisplatin [43], carboplatin and paclitaxel [41], avastin [32], doxorubicin [42], and sunitinib [44]. However, the only studies combining CA1P with radiation were our own preliminary reports with single treatments [24,45] or limited fractionated studies [24,25]. This lack of CA1P and radiation combination studies is surprising. Around 50% of cancer patients receive radiation therapy as part of their cancer treatment and many are treated with curative intent [46]. The vascular shut-down induced by the VDA potentially prevents oxygen and essential nutrients reaching cells downstream of the blockage, and cells already existing under conditions of low oxygen and nutrient levels (i.e., hypoxic cells), which are the major radiation resistant population, will likely die first. Those tumor cells remaining are likely the radiation-sensitive population.

However, sequencing of the two modalities could be a critical issue. With single-dose treatments, administering the VDA within a few hours after irradiating always produces an enhanced anti-tumor response [24], as also shown in Figure 2 and Figure 3. Whether this enhancement with CA1P is additive or synergistic is unclear and additional studies investigating this aspect should be undertaken. Treating mice with VDAs prior to irradiating generally has no benefit and in some instances can actually result in a reduced anti-tumor response compared to that found for radiation alone [24]. This suggests that some of the cells downstream of the vascular damage actually survive, despite becoming oxygen- and nutrient-deprived and, therefore, hypoxic and radiation-resistant. This has significant implications when trying to combine VDAs and fractionated radiation as in clinical practice.

CA1P does not show this schedule dependency (Figure 2 and Figure 3). This is probably a consequence of it having dual functional activity. Any cells made hypoxic due to the vascular damage are then killed by the cytotoxic action of the drug. Nevertheless, we recommend that the clinical application of this and other VDAs should be such that the drugs are administered after that radiation fraction where a sufficient time interval allows for any induced hypoxic cells to subsequently die, or reoxygenate, before the next radiation treatment is applied. That is why we chose a stereotactic schedule with 3–4-day intervals between each radiation (plus drug) treatment and administering the VDA after the last radiation treatment each week in the conventional radiation schedule, thus allowing a 3-day “weekend equivalent” interval before applying the next radiation treatment. With either of the stereotactic or conventional schedules, CA1P significantly enhanced tumor control (Figure 4 and Figure 5).

Our single-radiation and VDA dose schedules (Figure 2) indicated a plateau in the response with a VDA dose of 10–25 mg/kg. Increasing the dose substantially to a non-toxic dose of 100 mg/kg had no additional benefit. This dose dependency was also observed with the stereotactic schedules (Figure 4) and was suggested from the conventional radiation study (Figure 5). This implies a maximal benefit at doses way below the previously reported maximum tolerated dose of 250 mg/kg [21], which could be beneficial from a clinical standpoint where the use of low doses could avoid any potential systemic toxicity issues.

Of course, any benefits of combining radiation and VDAs, in terms of anti-tumor response, must be compared to radiation effects in dose-limiting normal tissues. We did not undertake such measurements using our stereotactic and conventional radiation studies, and this is something that should be done despite the complexity and difficulties in conducting such studies. However, we have previously published results for normal skin damage when single radiation and CA1P (50 mg/kg) treatments were combined, and although we found a small enhancement of radiation-induced early responding skin damage, this was less than that seen in tumors, so a therapeutic benefit was possible [45]. We have no reason not to expect similar findings when using stereotactic or conventional radiation approaches.

## 4. Materials and Methods

### 4.1. Animal and Tumor Model

All experiments were performed using a C3H mammary carcinoma that was implanted in the right rear foot of 10–14-week-old female CDF1 mice. This tumor model is an anaplastic adenocarcinoma that arose spontaneously in a C3H mouse at our institute and was originally designated as HB [47]; the name was changed to C3H mammary carcinoma when it was grown in the more stable CDF1 mouse variant [27]. C3H mammary carcinomas do not grow in culture; thus, experimental tumors were produced following sterile dissection of large flank tumors as previously described [24]. Basically, macroscopically viable tumor tissue was minced with scissors and 5–10 µL of this material implanted into the foot. Treatments were started when foot tumors reached a volume of 200 mm^3^, which generally occurred within 3 weeks after challenge; tumor volume was determined by the formula D1 × D2 × D3 × π/6 (where the D values represent three orthogonal diameters). Attempts were made to randomize the tumor-bearing mice into the different treatment groups. However, since the tumors grew at different rates, they did not achieve the 200 mm^3^ volume, when treatment was initiated, on the same day. Consequently, some selection was necessary to ensure that tumors beginning treatment on the same day were distributed among the different treatment groups.

### 4.2. Drug Preparation

CA1P was supplied by Oxigene Inc. (South San Francisco, CA, USA). It was freshly prepared before each experiment by dissolving in a sterile saline (0.9% NaCl) solution. Stock solutions were kept cold and protected from light. CA1P was injected intraperitoneally (i.p.) in a volume of 0.02 mL/g mouse body weight.

### 4.3. Radiation Treatments

Irradiations were given with a conventional therapeutic Philips X-ray machine, Koninklijke Philips N.V., Amsterdam, The Netherlands (230 kV; 15 mA; 2-mm Al filter; 1.1-mm Cu half-value layer; dose rate of 2.3 Gy/min). Dosimetry was accomplished by use of an integrating chamber. All treatments to tumor-bearing feet were administered to non-anesthetized mice placed in a specially constructed Lucite jig that restrained the animals, but allowed their tumor-bearing legs to be exposed and loosely attached to the jig with tape, as previously described [24]. Tumors only were irradiated, with the remainder of the animal being shielded by lead. To secure homogeneity of the radiation dose, tumors were immersed in a water bath with about 5 cm of water between the X-ray source and the tumor. The temperature of this water bath was maintained at 25 °C, which is the normal temperature of these foot tumors. For the stereotactic radiation treatments, mice were irradiated 3 times in a one-week period with a 3-to-4-day interval between each irradiation (i.e., Monday–Thursday–Monday or Tuesday–Friday–Tuesday). With the conventional radiation treatments, mice were irradiated twice daily (with a 6-h interval between each irradiation), 5 days a week (i.e., Monday to Friday), over a 3-week period.

### 4.4. Tumor Response Endpoints

The response of tumors to treatment was assessed using both a regrowth delay assay and local tumor control, as described previously [24]. For the regrowth delay assay, tumor volume was measured as described above on a daily basis following treatment. The tumor growth time (TGT5; time taken to reach five times the treatment volume) was then calculated from the regrowth curves. With the local tumor control assay, tumor-bearing animals were observed on a weekly basis up to 90 days post-treatment and the number of animals in each treatment group showing tumor control within this 90-day period was recorded. However, for the fractionated radiation treatments where repeated treatments with the same radiation dose were applied, dose–response curves were only possible using a clamped top-up dose; this approach ensured that the tumor microenvironment conditions were the same for all treatment groups. Basically, three days after the final stereotactic or conventional fractionation irradiation, mice had their tumor-bearing legs clamped with tubing and then treated with a range of radiation doses. The clamp was applied for 5 min before irradiating and maintained for the period of irradiation [24,25].

### 4.5. Statistical Analysis

For the tumor growth delay assay, the results are shown as the mean values ±1 SEM. Statistical analyses of these data were performed using Student’s *t*-test after testing for variance homogeneity using an F-test. The local tumor control data were subjected to logit analysis, and from the radiation dose–response curves the TCD50 value (radiation dose producing 50% tumor control) was calculated along with the 95% confidence intervals. Statistical analysis of these data was achieved using a chi-squared test. Regardless of the statistical test used, the level of significance was always *p* ≤ 0.05.

## 5. Conclusions

Numerous pre-clinical studies have clearly shown the potential benefit of combining VDAs with radiation to improve anti-tumor response, even though most of these studies were with single-dose drug and radiation treatments [15]. Our current pre-clinical study, using the most effective, non-toxic, VDA to yet be developed, clearly demonstrated how this particular agent should be applied with clinically relevant fractionated radiation treatments.

There are of course a number of potential limitations to our findings. Firstly, the results obtained were from only one tumor model and it would be nice to confirm the findings in other systems. Nevertheless, as stated previously, this C3H mammary carcinoma model has the advantage of allowing for tumor control assessment, which is the most clinically relevant endpoint for a local treatment such as radiation and most other rodent-based models do not permit such an analysis. Secondly, we only investigated one VDA, albeit the most potent agent for inducing vascular damage. It would be pertinent to undertake similar extensive fractionated radiation studies with other tubulin binding agents or agents from other classes of VDAs to determine whether the effects we report are in fact a universal finding.

Several VDAs have undergone clinical evaluation [18,48,49]; yet, despite the wealth of pre-clinical data showing the potential of VDAs to improve radiation therapy, only one study actually investigated a VDA/radiation combination [48]. Clearly, such a combination has the potential to significantly improve outcomes in cancer patients, and further pre-clinical and clinical studies should be encouraged.

## Figures and Tables

**Figure 1 ijms-25-08078-f001:**
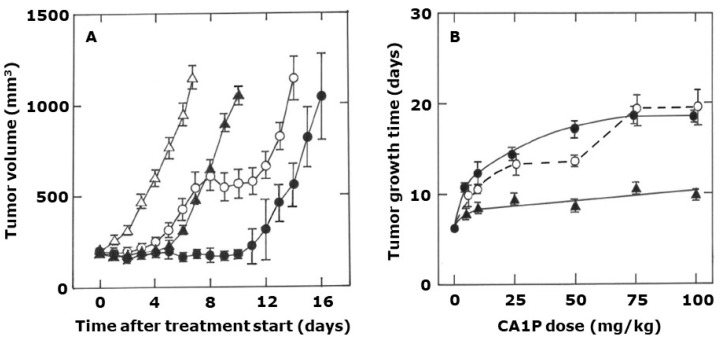
The schedule dependent effect of CA1P on tumor growth. (**A**) Tumor volume measured in untreated controls (**△**), or after the start of treatment with 50 mg/kg CA1P administered once only (▲), weekly (**⭘**), or three times in one week (⬤). (**B**) Tumor growth time when mice were treated with different CA1P doses administered either once (▲), weekly (**⭘**), or three times in one week (⬤). All results are means (±1 SEM) for an average of nine mice/group.

**Figure 2 ijms-25-08078-f002:**
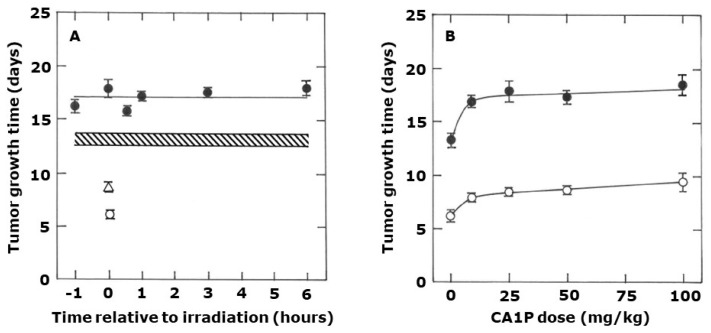
The effect of CA1P and radiation on tumor growth delay. (**A**) Tumor growth time in untreated controls (**⭘**), or after treatment with 50 mg/kg CA1P (**△**), 10 Gy radiation (shaded area), or radiation + CA1P (⬤). (**B**) Tumor growth time following treatment with different CA1P doses alone (**⭘**) or when CA1P injected 1 h after irradiating with 10 Gy radiation (⬤). Results are means (±1 SEM) for an average of nine mice/group.

**Figure 3 ijms-25-08078-f003:**
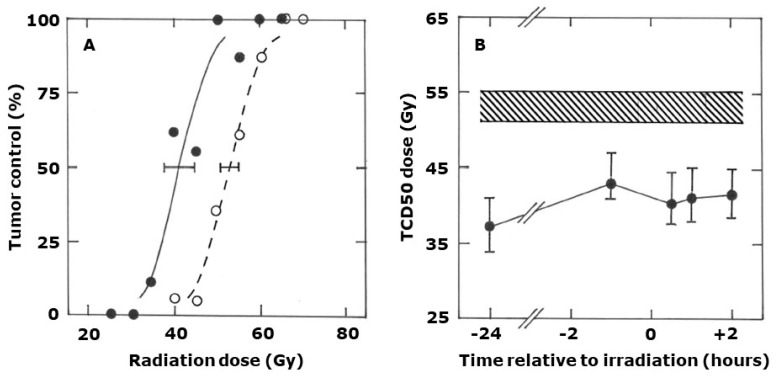
The effect of CA1P and radiation on local tumor control. (**A**) Percent local tumor control in mice treated with different single radiation doses (**⭘**) or when tumors were irradiated and then injected with CA1P (50 mg/kg) 30 min after irradiating (⬤). Results are for an average of nine mice/group with lines fitted following logit analysis. Errors are 95% confidence intervals at the TCD50 value. (**B**) TCD50 values (±95% confidence intervals) obtained from data like those shown in (**A**) for mice receiving radiation alone (shaded area) or when injected with CA1P (50 mg/kg) at various times before (−) or after (+) irradiating (⬤); radiation being applied at time 0.

**Figure 4 ijms-25-08078-f004:**
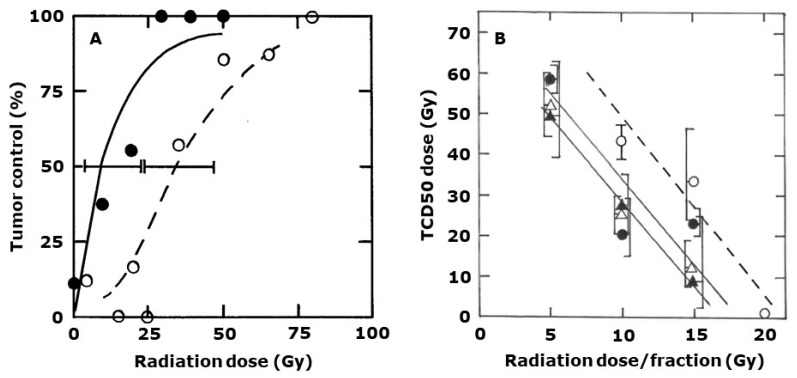
Combining CA1P with stereotactic radiation. (**A**) The percent local tumor control in mice initially treated with 3 × 15 Gy in one week (i.e., days 0, 4, 7) followed 3 days later with an additional radiation treatment, under clamped conditions, at the radiation doses indicated. Symbols are for radiation alone (**⭘**) or radiation with CA1P (10 mg/kg) injected 1 h after each of the 15 Gy fractions (⬤). Points represent the average of nine mice/group with the lines fitted following logit analysis. Errors are 95% confidence intervals at the TCD50 value. (**B**). TCD50 doses produced from data similar to those shown in (**A**), as a function of prior administration of 3 × 5–20 Gy. Symbols are for radiation alone (**⭘**), or radiation and CA1P at 5 mg/kg (⬤), 10 mg/kg (**△**), 15 mg/kg (▲). CA1P was injected 1 h after each of the three fractions. Errors on the data are 95% confidence intervals.

**Figure 5 ijms-25-08078-f005:**
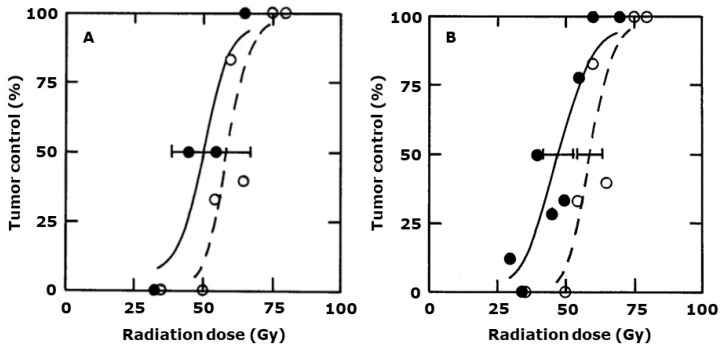
Combining CA1P with conventional fractionated radiation. (**A**) The percent local tumor control in mice treated with 30 × 2 Gy in a 3-week period (i.e., 2 fractions/day, 5 days/week) followed 3 days later with an additional radiation treatment, under clamped conditions, at the radiation doses indicated. Symbols are for radiation alone (**⭘**) or radiation with CA1P (10 mg/kg) injected 1 h after fractions 10, 20, and 30 (⬤). (**B**) Percent local tumor control following the clamped top-up doses in mice treated as described for (**A**), but where the symbols represent radiation alone (**⭘**) or radiation plus 25 mg/kg CA1P (⬤). Points represent the average of eight mice/group with the lines fitted following logit analysis. Errors are 95% confidence intervals at the TCD50 values.

## Data Availability

The data presented in this study are available on request from the corresponding author. All the data are stored in a central storage unit alongside data from all other animal experiments undertaken at our institute, thus making it impossible to allow separate access to any specific experiment.

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
