# Peer review of "Targeting the Tumor Vascular Supply to Enhance Radiation Therapy Administered in Single or Clinically Relevant Fractionated Schedules"

_ijms, 2024, doi:10.3390/ijms25158078_

Round 1

Reviewer 1 Report

Comments and Suggestions for Authors

The article is well planned, well designed study by the author. 
Introduction is well written with sufficient information. 
Materials and methods are well described. 
Results are clearly presented. 
Discussion is based on current literature and the results obtained. 
I would like to ask the author to add a separate conclusion and limitations of the study. 
Also a graphical abstract or illustration of the summary would be a good addition. 

Author Response

Comment 1: The article is well planned, well designed study by the author. 
Introduction is well written with sufficient information. 
Materials and methods are well described. 
Results are clearly presented. 
Discussion is based on current literature and the results obtained. 

Response 1: I thank the reviewer for their positive comments to all aspects of this manuscript. Glad to hear that no changes were requested for the various sections listed.

Comment 2:  I would like to ask the author to add a separate conclusion and limitations of the study. 

Response 2: This was a good suggestion by the reviewer and we have now addressed this issue by adding a separate "Conclusions" section on page 7. It also includes a paragraph of the potential limitations of our study (see lines 241-250). 

Comment 3: Also a graphical abstract or illustration of the summary would be a good addition. 

Response 3: At the moment this has not been included. I can understand the reviewer's idea behind such an addition. However, with more than 40 years experience as a researcher and with more than 300 publications, this is the first time I have been asked to create a Graphical Abstract, so have no experience with such things. Consequently, I have spent a lot of time trying to create such a set of images for this study and nothing seems to be working. What I have is either too ridiculous, too complicated, or simply does not represent the study. It's going to be a long time before I achieve an adequate Graphical Abstract.   

Reviewer 2 Report

Comments and Suggestions for Authors

The manuscript "Targeting the tumor vascular supply to enhance radiation therapy administered in single or clinically relevant fractionated schedules" reports on the pre-clinical studies of using combined therapy in C3H mammary carcinoma.  The authors have chosen a VDA agent combretastatin -1 phosphate (CA1P) and radiotherapy, the studies being reported for the first time. The authors also compared their results with other similar drug CA4P.

The Introduction section clear highlighted the state-of-the-art on using the VDA drugs, their intrinsic mechanism of action and motivated their studies by evaluating the dose and time dependent effects of VDA with radiation to inhibit the tumor growth.

The Results and Discussion sections are clearly presented, the figures and explanations are deeply investigated, the originality of the results is often mentioned and the comparisons with other studies are cited.

The Conclusion section is supported by the results! This paper deserves to be published as the results are of great importance and uniqueness!

Author Response

Comment 1: 

The manuscript "Targeting the tumor vascular supply to enhance radiation therapy administered in single or clinically relevant fractionated schedules" reports on the pre-clinical studies of using combined therapy in C3H mammary carcinoma.  The authors have chosen a VDA agent combretastatin -1 phosphate (CA1P) and radiotherapy, the studies being reported for the first time. The authors also compared their results with other similar drug CA4P.

The Introduction section clear highlighted the state-of-the-art on using the VDA drugs, their intrinsic mechanism of action and motivated their studies by evaluating the dose and time dependent effects of VDA with radiation to inhibit the tumor growth.

The Results and Discussion sections are clearly presented, the figures and explanations are deeply investigated, the originality of the results is often mentioned and the comparisons with other studies are cited.

The Conclusion section is supported by the results! This paper deserves to be published as the results are of great importance and uniqueness!

Response 1: I thank the reviewer for such positive comments to this study and for not requesting any changes.